# First Cases of Feline Sporotrichosis Caused by *Sporothrix brasiliensis* in Paraguay

**DOI:** 10.3390/jof9100972

**Published:** 2023-09-27

**Authors:** Carolina Melchior do Prado, Emanuel Razzolini, Gabriela Santacruz, Leticia Ojeda, Marlon Roger Geraldo, Nancy Segovia, José Pereira Brunelli, Vânia Aparecida Vicente, Walfrido Kühl Svoboda, Flávio Queiroz-Telles

**Affiliations:** 1Postgraduate Program in Microbiology, Parasitology and Pathology, Biological Sciences, Department of Basic Pathology, Federal University of Parana, Curitiba 81531-980, Brazil; c.melchior.mv@gmail.com (C.M.d.P.); marlonportaldofuturo@gmail.com (M.R.G.); vaniava63@gmail.com (V.A.V.); 2Postgraduate Program in Bioprocess and Biotechnology Engineering, Technology Sector, Department of Biotechnology, Federal University of Parana, Curitiba 81531-990, Brazil; erazzolini@gmail.com; 3Regional Epidemiological Laboratory, Faculty of Health Sciences, National University of the East, Minga Guazú 7420, Paraguay; mgsantcruz@gmail.com (G.S.); letiojeda@hotmail.com (L.O.); dranancysegovia@gmail.com (N.S.); 4Ministry of Public Health and Social Welfare, Asuncion 2160, Paraguay; jose_pereira15@hotmail.com; 5Department of Public Health, Federal University of Latin American Integration, Foz do Iguaçu 85870-650, Brazil; walfrido.ufpr@gmail.com; 6Department of Public Health, Hospital de Clínicas, Federal University of Paraná, Curitiba 80060-900, Brazil

**Keywords:** sporotrichosis, *Sporothrix brasiliensis* infection, diagnosis, zoonotic infectious diseases

## Abstract

*Sporothrix brasiliensis* is an emerging fungal pathogen causing cat-transmitted sporotrichosis, an epi-zoonosis affecting humans, cats and dogs in Brazil and now spreading to neighboring South American countries. Here, we report the first two autochthonous cases of cat-transmitted sporotrichosis in Paraguay. The first case was a four-year-old male cat showing several ulcerative lesions, nasal deformity and respiratory symptoms. The second case was a one-year-old male cat showing a single ulcerated lesion, respiratory symptoms and nasal deformity. Both cases were admitted to a veterinary clinic in Ciudad del Este, Paraguay. Isolates were recovered from swabs of the two cases. Using molecular methods, the isolates were identified as *S. brasiliensis*.

## 1. Introduction

Sporotrichosis, caused by fungi of the genus *Sporothrix,* is the most prevalent and globally distributed of the implantation mycoses [1]. Transmission is usually by transcutaneous or transmucosal traumatic inoculation of the fungus present in soil, plants and decomposing organic matter (sapronotic), and through enzootic (cat-to-cat/dog) and zoonotic (cat-to-human) routes. *Sporothrix schenckii* and *Sporothrix brasiliensis* are the most common etiological agents in human and animal diseases in Brazil [2,3].

Cat-transmitted sporotrichosis (CTS) is a neglected disease. Since the 1990s, this important zoonosis has emerged as a public health problem in Brazil, where it has been responsible for thousands of associated human and feline cases, which almost exclusively involve infections with *S. brasiliensis* [4]. Cases have quickly spread throughout Brazil and to other countries in South America [5]. In the enzootic (cat-to-cat, cat-to-dog) and zoonotic (cat-to-human) routes, transmission can occur through bites, scratches, contact with exudate from skin lesions and probably through respiratory droplets during cat sneezing [6,7].

In cats, clinical manifestations may include single or multiple ulcerative lesions on the skin, associated with enlarged lymph nodes and respiratory signs (mainly sneezing). Most commonly, cats can present a disseminated disease with multiple cutaneous lesions and mucosal involvement, especially the nasal mucosa. Conjunctival, oral and genital mucosa may also be affected [8]. Culture isolation is the gold standard technique for the diagnosis of sporotrichosis in animals. But since infected cats display a high fungal burden, the direct examination of yeast cells isolated from skin lesions is frequently used by veterinarians and microbiologists in clinical practice [6]. Molecular identification methods based on the sequence analysis of the calmodulin gene are required to identify the *Sporothrix* species [9].

In Paraguay, the only reported CTS cases so far occurred in 2017 among relatives who moved from Brazil with an infected cat. Laboratory tests performed identified the etiological agent as *Sporothrix* spp. (without molecular identification). This was the first record of a travel-associated case of *Sporothrix* outside Brazil or Argentina, documenting the potential for spread to new areas through the transport of infected cats [10]. Other cases supporting CTS’s potential to spread were observed in the U.S. and U.K. In 2020, a Brazilian woman was diagnosed with CTS in Boston, U.S., after being bitten by a sick cat in Minas Gerais, Brazil [11]. More recently, researchers reported the first human and feline *S. brasiliensis* cases in the U.K., both involving people who had moved from Brazil [12]. In 2019, the World Health Organization published an international alert on the potential risk of transmission of infection by *S. brasiliensis* in neighboring countries [13]. Besides the spread of CTS in Brazil, sporotrichosis cases caused by *S. brasiliensis* have been observed in cats in other Latin American countries without previous exposure to Brazil [14,15]. In Argentina, for example, *S. brasiliensis* has already been described in cats from the provinces of Buenos Aires and Santa Cruz, located in the Argentinean Patagonian region. In Misiones, a province bordering Brazil, there was a single description of *S. brasiliensis* isolated from a human in 1986 [15]. In 2022, researchers from Chile described three cases of domestic and feral cats with *S. brasiliensis* in the Magallanes region of southern Chile. Those cases were the first isolation reports of *S. brasiliensis* in cats from Chile. Cases 1 and 2 came from the Río Verde County, a small rural village, and case 3 came from the city of Punta Arenas [14]. Here, we report the first autochthonous feline cases of CTS by *S. brasiliensis* in Paraguay (PY).

## 2. Case Presentation

This study was approved by the Committee for Ethics in Research of the Federal University of Parana (number—CAAE 52726021.8.0000.0102) and by the Animal Use Ethics Committee of the Agricultural Sciences Campus of the Federal University of Parana.

Ciudad del Este (CDE) (25°29′36″ S and 54°40′18″ W) is a city and district of PY, located in the extreme east of the country on the banks of the Paraná River. More specifically, it is situated on the Triple Border with Argentina and Brazil and has approximately 300,000 inhabitants. It is the capital of the Alto Paraná department, located 327 km from the country’s capital—Asunción.

The first case observed in this city was a four-year-old mixed-breed male cat admitted on June 22 to a veterinary clinic in CDE, with numerous ulcerated lesions and respiratory symptoms. The animal had a history of respiratory signs (sneezing, nasal secretion and respiratory difficulty) for about twenty-one days and ulcerated lesions for about two months (beginning with lesions on the tail and later involving lesions on the face and paws). After a physical examination, four lesions in the cranial region (Figure 1A), a small nasal deformity (Figure 1A), one lesion in the left hind limb (Figure 1B) and four ulcerated lesions in the initial and medial portion of the tail (which had been amputated due to tissue necrosis) were found (Figure 1C).

Two months later (on August 22), a second case was admitted to the same veterinary clinic in CDE. It was a one-year-old mixed-breed male cat showing a single ulcerated lesion in the nose, respiratory symptoms (sneezing, nasal secretion and respiratory difficulty) and nasal deformation. The onset of clinical signs occurred eight days before the presentation at the clinic. After a physical examination, the only ulcerated lesion was confirmed, together with a significant nasal deformity (Figure 1D). This patient, who presented symptoms two months after the diagnosis of the first case, was from the same residence as the previous one.

In both cases, feline sporotrichosis was determined as the main diagnostic suspicion. Relevant clinical and epidemiological data were collected through clinical–epidemiological questionnaires applied to the owners (Figure A1 and Figure A2). The animals had a history of climbing and scratching trees and burying their feces outside. They were neutered and had no history of other previous illnesses. Both had had only their first rabies vaccine and access to the yard, the street and other residences. The owners reported that both animals had always lived in CDE and had no previous travel history. The geographic coordinates of the cats’ residence were established using Google Earth^®^ software. Cartographic bases from the Brazilian Institute of Geography and Statistics (IBGE) were used for georeferencing the coordinates. Quantus Gis software (QGIS) was used to assemble the map (Figure 2). The animals had not received any previous treatment with antibiotics and antifungals.

In the first case, the patient was referred for euthanasia due to his poorly clinical status. In the second case, 100 mg/day itraconazole and 30 mg/day potassium iodide therapy were applied for about eight months until the clinical signs disappeared and were maintained for another two months to minimize the risk of recurrence.

For both cases, swab samples of the lesions and slides with an imprint of the lesions were collected to perform the definitive diagnosis. Mycological examination of the slide samples was performed based on direct Gram-stain microscopy, which enabled visualizing cigar-shaped yeast structures in the feline samples (Figure 3A). Considering that culture is the gold standard for the diagnosis of sporotrichosis [6], swabs collected from the wounds were placed in Stuart’s medium and sent to the laboratory for mycological cultures. The samples were cultivated on Sabouraud dextrose agar (SDA) (KASVI, ttps://www.kasvi.com.br, accessed on 9 June 2022) containing chloramphenicol and incubated at 25 °C for the growth of colonies for up to 10 days.

Macroscopic and microscopic characteristics of the colonies were observed and confirmed the fungi of the genus *Sporothrix* identity. The clinical samples showed growth of a white, creamy colony at 25 °C (Figure 3B). For the identification of the mycelial phase, monosporic cultures were grown on SDA at room temperature. We used the slide culture method to characterize microscopic filamentous features of the *Sporothrix* spp. isolates. We inoculated mycelia fragments into 1 × 1 cm SDA blocks, which were then mounted on a slide, covered with a coverslip and incubated for 14–21 days at room temperature. The mycelial preparation was stained with lactophenol blue and visualized with a Zeiss AxioObserver Z1 microscope equipped with a 40× objective. The microscopical analysis showed hyaline hyphae and septate with rounded unicellular conidia (Figure 3C).

Molecular analysis of the two isolates was based on calmodulin (CAL) gene sequences. For molecular identification, colonies cultivated on Sabouraud glucose agar (SGA) were used for DNA extraction. The fungi mycelium was transferred to a 1.5 mL tube containing 100 mg of a silica celite mixture (2:1, *w*/*w*) and 300 µL CTAB buffer [CTAB 2% (*w*/*v*), NaCl 1.4 M, Tris-HCl 100 mM, pH 8.0; EDTA 20 mM]. Fungi cells were disrupted with a pestle for 5 min and incubated for 40 min at 65 °C. Then, 500 µL 24:1 chloroform: isoamyl alcohol (CIA) was added and the solution was centrifuged for 10 min at 20,500× *g*. The supernatant was collected in a new tube with 2 vols of ice-cold 96% ethanol. The DNA was precipitated for 2 h at −20 °C and then centrifuged for 10 min at 20,500× *g*, washed with cold 70% ethanol, dried at room temperature and resuspended in 100 µL in ultrapure water [16]. DNA integrity was assessed by running a 5 μL aliquot of the extraction on a 0.8% agar agarose-gel electrophoresis stained with 0.5 μg/ mL ethidium bromide. To estimate the DNA yield, we used a NanoDrop 1000c Instrument (Thermo Fisher Scientific, Waltham, MA, USA). Amplification of the CAL was performed using CL1 and CL2A primers [17]. PCR reaction mixtures were made with a total volume of 12.5 μL (1× PCR buffer, 2.0 mM MgCl2, 25 μM deoxynucleotide triphosphates (dNTPs)), 0.5 μM of each forward and reverse primer, 1 U of Taq DNA polymerase (Ludwig Biotec, Bela Vista, Brazil) and 10 ng of genomic DNA. The PCR parameters for amplification were 94 °C for 3 min, followed by 35 cycles consisting of 94 °C for 35 s, 58 °C for 30 s and 72 °C for 1 min, and a delay at 72 °C for 1 min, performed in an ABI Prism 2720 thermocycler (Applied Biosystems, Foster City, CA, USA). Amplicons were sequenced with BigDye Terminator cycle sequencing kit v. 3.1 (Applied Biosystems, Foster City, CA, USA) according to the manufacturer’s instructions, using the same primers of the PCR, and the amplification conditions were as follows: 95 °C for 1 min, followed by 30 cycles consisting of 95 °C for 10 s, 50 °C for 5 s and 60 °C. The consensus sequence of the CAL regions was visually inspected using MEGA v.7 software [18] and compared to the GenBank Blast (NCBI). The sequences of CAL were aligned with reference strains (Table A1) using the online MUSCLE interface and the substitution model was selected for each genus using the MEGA software (GTR + G). The chosen index was the one that presented the lowest AIC. The phylogeny was constructed based on the CAL gene. Maximum likelihood analysis was performed in the RaxML-HPC2 [19] implemented in the CIPRES server using the 1000 bootstrap to reach split frequencies of ≤0.01. The resulting trees were plotted in FigTree v.1.4.2 [20]. The results showed that the clinical strains were *Sporothrix brasiliensis* (Figure 4).

## 3. Discussion

These are the first feline cases of CTS in PY. Human sporotrichosis is a deep mycosis, relatively common in PY. In PY, this disease is still mostly related to the classic form of transmission, the sapronotic route, and has the greatest effect on certain professions such as carpenters, gardeners and farmers [10].

Ciudad del Este is situated near the Iguazu Falls at the busiest frontier in Brazil and in the context of the triple international border between Brazil, Paraguay and Argentina. The flow of people, animals, plants and microorganisms, sometimes without proper inspection and monitoring, is constant. And in terms of the flow of people, Foz do Iguaçu, as an important tourist destination, presents other peculiarities, occupying the third place in the preference of foreign tourists arriving in the country. Therefore, it is necessary to reflect on how these international borders can contribute to the dissemination of microorganisms such as *S. brasiliensis* in view of the lack of specific public health policies in this region [21].

In recent decades, the domestic cat (*Felis catus*) has played an important role in the epidemiology of sporotrichosis due to its increased susceptibility to this disease and its role as a vehicle of transmission to humans through bites, scratches, sneezing and contact with secretions from infected animals [8]. Sporotrichosis in cats is generally systemic, showing disseminated lesions with a high burden of infectious yeast-like cells and which, without treatment, cause death [22]. Infected animals show disseminated cutaneous lesions characterized by ulcers and nodules with exudate, especially on the face. Upon licking or touching with the extremities, the agent is transferred to the paws or the mouth, facilitating contagion. This behavior, together with the high burden of mycotic elements in the wounds, facilitates contagion among cats and other animals such as dogs and humans [23].

Although the first animal was euthanized, it is important to note that sporotrichosis is treatable. The drugs indicated for sporotrichosis treatment are itraconazole, potassium iodide, terbinafine and amphotericin B. Itraconazole is the first choice for treatment, at a dose of 100 mg/day until the complete resolution of the lesions [24]. Treatment duration varies in the literature and is determined by the patient’s clinical response. Furthermore, it is important to point out that the capacity for transmission by the infected cat reduces exponentially with the implementation of the treatment [15].

Furthermore, the animal’s history of access to the streets should always be considered during the period of treatment, especially in endemic regions [23]. Therefore, it is worth highlighting the importance of considering sporotrichosis as a differential diagnosis for animals showing skin lesions due to the potential zoonotic risk of *S. brasiliensis* transmission [25].

Animals under suspicion of sporotrichosis should be isolated from other animals. Decontamination of the environment can be carried out with sodium hypochlorite (bleach) [8]. Injuries resulting from the typical behavior of cats in distress should be avoided through the application of correct animal restraint techniques and the use of personal protective equipment, such as procedure gloves, aprons and goggles [26]. This is especially important as this fungal species has a high transmission potential due to contact with secretions and exudates from the lesions of sick cats during treatment and handling procedures. Feline management methods based on the “Cat-Friendly Practices” philosophy can help prevent injuries, as they aim to keep the patient calm so that their handling is peaceful and safe [27].

## Figures and Tables

**Figure 1 jof-09-00972-f001:**
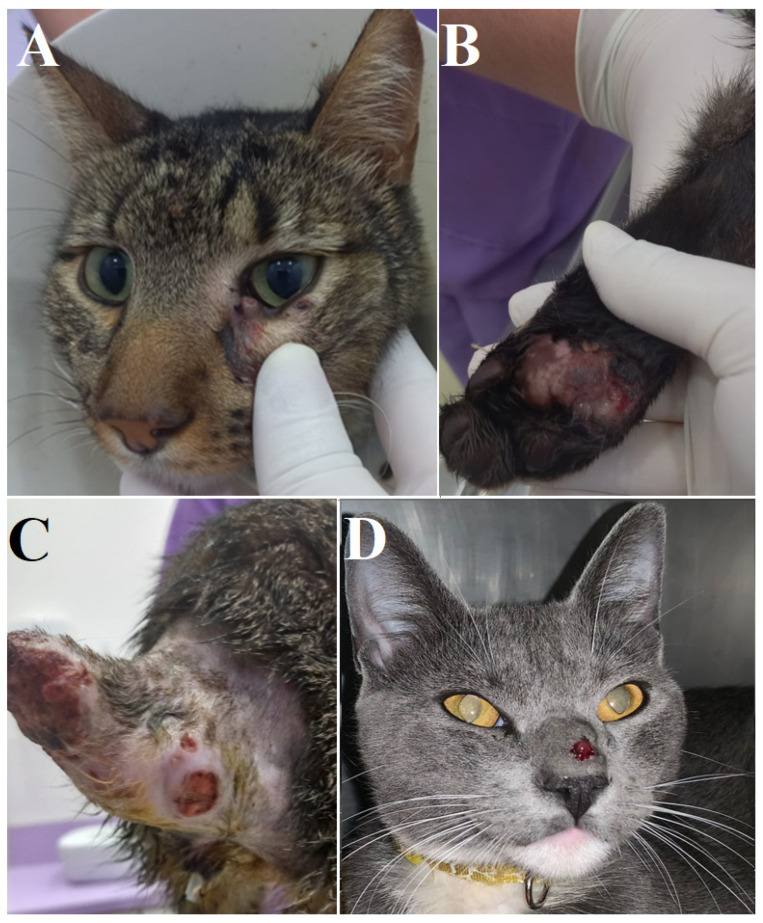
Lesions in the cranial region and a small nasal deformity (**A**), ulcerated lesion in the left hind limb (**B**) and ulcerated lesions in the initial and medial portion of the tail (**C**) found after physical examination of animal 1. Nasal deformity and ulcerated lesion (**D**) found after physical examination of the animal 2.

**Figure 2 jof-09-00972-f002:**
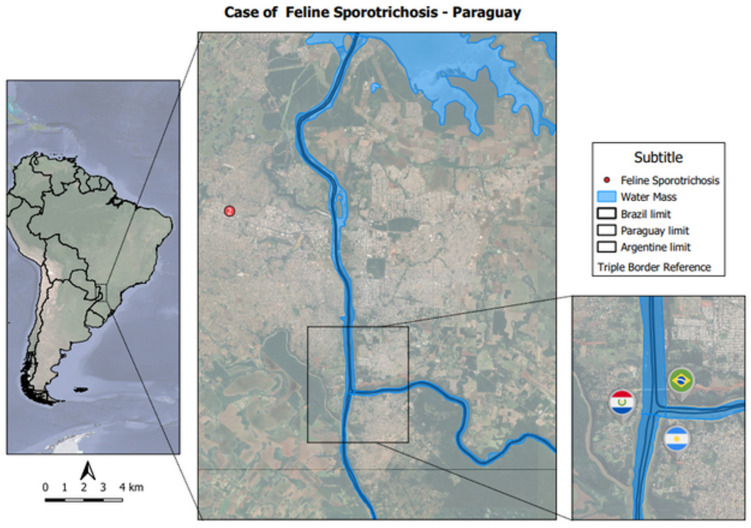
Spatial distribution of the two animals with a positive diagnosis for sporotrichosis in Ciudad del Este, Paraguay.

**Figure 3 jof-09-00972-f003:**
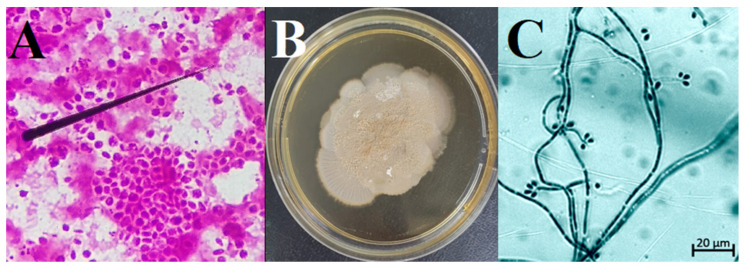
(**A**) Cigar-shaped yeast structures (black arrow) observed in feline samples (Gram-stain, 1000×). (**B**) *Sporothrix* colony after growth on Sabouraud agar with chloramphenicol. (**C**) Micromorphology of *Sporothrix* spp. colonies (black arrow) (lactophenol cotton blue ×400).

**Figure 4 jof-09-00972-f004:**
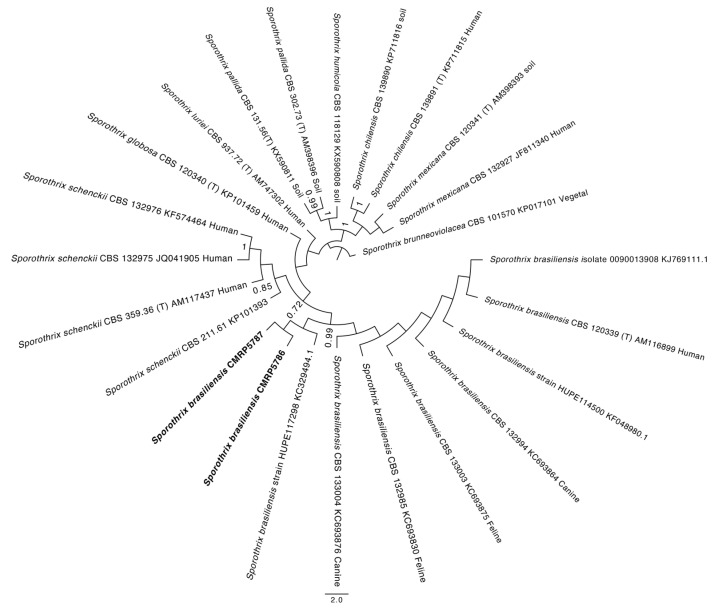
Phylogenetic tree of *Sporothrix brasiliensis*, obtained by ML analysis based on partial sequences of the CAL. The isolates CMRP5786 and CMRP5787 represent the isolates of Paraguay. Reference strains used can be seen in Table A1.

## Data Availability

The data presented in this study are openly available in NCBI, accession number OR501574 and OR501573.

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
