# Peer review of "First Cases of Feline Sporotrichosis Caused by Sporothrix brasiliensis in Paraguay"

_jof, 2023, doi:10.3390/jof9100972_

Round 1

Reviewer 1 Report

Epidemiological surveillance is relevant, and including molecular evidence is essential. However, the work needs to be improved in some aspects:

1)    The manuscript needs more information about the sequences used for constructing the phylogenetic tree, improved image sharpness, and improved sharpness in the bootstrap values.

2)    GenBank accession numbers of the generated sequences are recommended.

3)    The CAL locus is a good marker to differentiate Sporothrix species and has been used in several studies involving this fungus. Although CAL can differentiate species, it cannot provide sufficient data about intraspecific genetic diversity for S. brasiliensis and S. globose. BT2, EF-1a and CHS1 are also widely used to effectively identify sporotrichosis agents (de Carvalho et al. 2022). 

4)    It is recommended to include distance-matrix methods (UPGMA/Neighbor-joining) and Bayesian inference to compare topologies, and clarify how the substitution model was determined.

5)    Laboratory and blood investigations should be obtained to deepen the case (comorbidities).

No comments

Author Response

Response to Reviewer X Comments

1. Summary

2. Questions for General Evaluation

Reviewer’s Evaluation

Response and Revisions

Does the introduction provide sufficient background and include all relevant references?

Yes.

Are all the cited references relevant to the research?

Yes.

Is the research design appropriate?

Must be improved.

Are the methods adequately described?

Must be improved.

Are the results clearly presented?

Must be improved.

Are the conclusions supported by the results?

Must be improved.

3. Point-by-point response to Comments and Suggestions for Authors

Comments 1: The manuscript needs more information about the sequences used for constructing the phylogenetic tree, improved image sharpness, and improved sharpness in the bootstrap values.

Response 1: Thank you for pointing this out. We agree with this comment. We added Table 1A with relevant data about the sequences used for constructing the phylogenetic tree. We also improved image sharpness, and improved sharpness in the bootstrap values. These changes can be found in pages 6, 8, 9, 10 and 11.

Comments 2: GenBank accession numbers of the generated sequences are recommended.

Response 2: Thank you for pointing this out. We agree with this comment. We put those information’s in the Data Availability Statement and in Table 1A the GenBank accession numbers of the generated sequences. These changes can be found in pages 7, 10 and 11.

Comments 3: The CAL locus is a good marker to differentiate Sporothrix species and has been used in several studies involving this fungus. Although CAL can differentiate species, it cannot provide sufficient data about intraspecific genetic diversity for S. brasiliensis and S. globose. BT2, EF-1a and CHS1 are also widely used to effectively identify sporotrichosis agents (de Carvalho et al. 2022).

Response 3: Thank you for pointing this out. We agree with this comment. But we used calmodulin because the objective of the work was not to analyze the diversity of the species but rather to make a taxonomic classification. However, we know that other genes are necessary for studies of intraspecies variability and diversity. In future studies, if such analyzes are carried out, the recommended genes will be used.

Comments 4: It is recommended to include distance-matrix methods (UPGMA/Neighbor-joining) and Bayesian inference to compare topologies, and clarify how the substitution model was determined.

Response 4: Thank you for pointing this out. We agree with this comment. We have included a distance-matrix method in the supplementary material.

We chose to use the maximum likelihood (ML) approach because the main objective of the work was simply to separate species into a phylogenetic tree. In these cases, the ML approach is simpler to implement and requires fewer parameter adjustments compared to Bayesian inference. Furthermore, ML is generally more computationally efficient than Bayesian inference. ML also does not require specifying prior distributions for model parameters, unlike Bayesian inference, which relies on prior distributions. ML is more robust to poor prior choices, which can be a problem in Bayesian inference if prior distributions are not well informed. ML has been widely used in phylogeny and is considered the standard approach for constructing phylogenetic trees. Many tools and software are available to perform ML analysis.

Comments 5: Laboratory and blood investigations should be obtained to deepen the case (comorbidities).

Response 5: Unfortunately, blood investigations were not possible.

Reviewer 2 Report

The work is generally carried out adequately and has clinical relevance since it is the first report in that region on mycosis; which can be quickly distributed if proper precautions are not taken.

Some considerations to take into account to improve the research work:

Lines 29-30: references are needed 

line 33: Full name of Sporothrix schenckii and Sporothrix brasiliensis

Line 35-37: references are needed

line 151-158: References of this method are needed

Fig 3. If it is possible to improve the image of panel A, place the metric scale. If you can change it, to be able to observe the fungal morphology properly.

Round 2

Reviewer 1 Report

1. Title should be changed to "First cases of feline sporotrichosis caused by Sporothrix brasiliensis in Paraguay"

2. Eliminate sequences of the phylogenetic tree that do not correspond to studied cases from this study

3. Clarify the selection of the substitution model for the construction of the phylogenetic tree.

4. The GenBank accession numbers are not available yet.
